# Overexpression of *GmbZIP59* Confers Broad-Spectrum Stress Resistance in *Arabidopsis thaliana* and Rice (*Oryza sativa*)

**DOI:** 10.3390/plants14213326

**Published:** 2025-10-30

**Authors:** Mengnan Chai, Tingyu Liu, Xunlian Fang, Danlin Dou, Zhuangyuan Cao, Ziqi Liu, Xiaoyuan Xu, Simin Ma, Kangmin Zhu, Lian Yu, Yuan Qin, Maokai Yan, Hanyang Cai

**Affiliations:** College of Life Sciences, Fujian Provincial Key Laboratory of Haixia Applied Plant Systems Biology, State Key Laboratory of Ecological Pest Control for Fujian and Taiwan Crops, Fujian Agriculture and Forestry University, Fuzhou 350000, China

**Keywords:** soybean, bZIP transcription factor, *GmbZIP59*, salt, drought

## Abstract

Soybean (*Glycine max*) is a vital oilseed and economic crop in China, often constrained by drought, salinity, and biotic stresses. In this study, we identified a soybean bZIP transcription factor, *GmbZIP59*, whose expression is upregulated by salt, drought, ethylene (ETH), methyl jasmonate (MeJA), and abscisic acid (ABA). Overexpression of *GmbZIP59* in *Arabidopsis* (*OE-13* and *OE-20*, two independent *Arabidopsis* transgenic lines) exhibited enhanced resistance to *Sclerotinia sclerotiorum* (*S. sclerotiorum*), improved tolerance to salt stress, and increased sensitivity to phytohormones. Overexpression of *GmbZIP59* in rice (*OE-1* and *OE-2*, two independent rice transgenic lines) improved tolerance to salt and drought stresses. Quantitative real-time polymerase chain reaction (qRT-PCR) analysis revealed that elevated expression of stress-related genes occurred in transgenic lines under adverse conditions. Furthermore, chromatin immunoprecipitation-qPCR (ChIP-qPCR) assays confirmed that GmbZIP59 directly binds to the promoters of ETH, MeJA, and ABA, responsive genes associated with stress responses. These findings demonstrate that *GmbZIP59* acts as a positive regulator of biotic and abiotic stress tolerance in soybean.

## 1. Introduction

As sessile organisms, plants lack the capacity to evade adverse environmental conditions, unlike motile animals. Plants are subjected to various biotic and abiotic stresses, such as salinity, drought, and extreme temperatures, significantly affecting their growth and development processes [1]. In response, plants have developed highly sophisticated systems, encompassing stress signal detection, transduction, and subsequent physiological and biochemical adaptations [2,3]. Critically, the efficacy of these responses is governed by the precise temporal dynamics of hormonal signaling networks, which coordinate defense and developmental programs over time [4].

Transcription factors (TFs) are key molecular switches in these mechanisms, orchestrating stress signal transduction and fine-tuning downstream gene expression to protect plants from stress-related damage [5,6,7]. bZIP transcription factors function as central regulators in plant signaling networks, coordinating immune responses and adaptation to biotic and abiotic stresses [8]. The bZIP transcription factor family possesses a characteristic bZIP domain [9,10,11]. This domain, typically spanning 40–80 amino acids, comprises two functionally distinct α-helix subregions: (1) a basic region for DNA-binding and (2) an adjacent leucine zipper motif mediating protein dimerization [10,12,13].

The bZIP TFs are involved in various biological processes, ranging from developing flowers, seeds, and roots in plants to mediating responses to different biotic and abiotic stresses, including salt, drought, low temperatures, and pathogen attacks [14,15]. For instance, in *Arabidopsis*, the bZIP transcription factor *HY5* regulates *AtERF11*, a key repressor of ethylene biosynthesis [16]. Meanwhile, in soybeans, *GmbZIP5* is an additional cofactor for GmMYB176 to regulate isoflavonoid biosynthesis [17]. Functional studies further demonstrate that *GmbZIP152* responds to both biotic and abiotic stresses in soybean by activating the expression of antioxidant enzyme-related genes and by directly binding to the promoters of stress-responsive genes [18]. *GmbZIP60*, a positive regulator in soybean, enhances salt and drought stress tolerance in transgenic soybean and rice via binding to the promoters of abiotic stress-related genes [19]. Conversely, *GmbZIP71-4* functions as a negative regulator in waterlogging tolerance by binding to the promoter of the *GmABF2* gene and mediating the plant’s tolerance to flood stress through the abscisic acid signaling pathway [20].

In our previous study, *GmbZIP59* was demonstrated to have high expression under drought and waterlogging stress conditions through transcriptomic and RT-qPCR analyses [21]. This study provides the first comprehensive functional characterization of this novel transcription factor. We successfully isolated and cloned the *GmbZIP59* gene from soybean. Expression profiling demonstrated that *GmbZIP59* is significantly upregulated in response to *S. sclerotiorum* infection, salt stress, and drought stress. Transgenic plants containing *35S::GmbZIP59::GFP* in *Arabidopsis* and rice exhibited enhanced resistance to *S. sclerotiorum* and improved tolerance to salt and drought stresses compared to wild-type (WT). These findings reveal a previously uncharacterized role of *GmbZIP59* as a positive regulator in plant defense responses against diverse biotic (*S. sclerotiorum*) and abiotic (salt and drought) stresses. We hypothesize that *GmbZIP59* functions as a cross-regulator integrating multiple stress signaling pathways. This represents the first report on the *GmbZIP59* transcription factor conferring such dual resistance, providing critical theoretical insights for plant genetics and developmental biology while offering potential applications for soybean cultivar enhancement and yield improvement.

## 2. Results

### 2.1. Structural Characterization and Domain Architecture of the GmbZIP59 Transcription Factor

*GmbZIP59* cDNA consists of 1273 bp (Appendix A) and encodes a bZIP-structured protein (Appendix A) with a relative molecular mass of 16.96 kDa and a theoretical isoelectric point (pI) of 5.3. We retrieved the amino-acid sequence of *GmbZIP59* (Glyma. 10G013300) and its orthologous protein from the Phytozome 14 database. Comprehensive alignments demonstrated that *GmbZIP59* exhibits pronounced sequence conservation relative to *GmbZIP60* (Glyma.02G012700) from soybean (*Glycine max*), *AtbZIP53* (AT3G62420) and *AtbZIP44* (AT1G75390) from *Arabidopsis* (*Arabidopsis thaliana*), *OsbZIP38* (LOC_Os05g03860) from rice (*Oryza sativa*). Sequence inspection revealed that *GmbZIP59*, together with its homologues, harbors a nuclear-localization signal (NLS), a conserved bZIP DNA-binding domain, and a leucine zipper dimerization motif (Appendix A).

The sequence logo analysis of GmbZIP59 revealed a highly conserved bZIP domain, featuring the characteristic basic region and leucine zipper structure. Phylogenetic analysis identified AtbZIP53 and AtbZIP44 as the closest Arabidopsis orthologs, sharing 48.82% and 45.45% overall sequence identity with GmbZIP59. Notably, the conservation within the DNA-binding basic region is significantly higher, explaining the similar DNA-binding specificity observed between these transcription factors (Appendix A). Furthermore, in silico analysis of the GmbZIP59 promoter (up to 2.0 kb upstream) identified several stress-related cis-acting elements, including ABA-, MeJA-, salicylic acid (SA)-, and dehydration-responsive motifs (Appendix A). Collectively, these findings clarify the evolutionary relationship of GmbZIP59 within the bZIP family and strongly support its potential role in mediating diverse stress response pathways.

### 2.2. Stress-Responsive Expression Dynamics of the GmbZIP59 Transcription Factor in Soybean

Two-week-old soybean (*Glycine max*) seedlings were exposed to multiple stress treatments, with leaf samples collected at 6, 12, and 24 h post-treatment for RNA extraction and subsequent RT-qPCR analysis of *GmbZIP59* expression level (Figure 1). Quantitative analysis revealed that *GmbZIP59* expression was significantly modulated in response to multiple stresses, including infection by *S. sclerotiorum*, as well as treatments with salt, mannitol, ETH, MeJA, and ABA. Following salt (Figure 1B), mannitol (Figure 1C), and ETH (Figure 1D) treatments, *GmbZIP59* transcript levels showed rapid induction, peaking at 6 h post-treatment before declining. In contrast, exposure to both *S. sclerotiorum* infection (Figure 1A) and MeJA (Figure 1E) triggered progressive *GmbZIP59* accumulation, reaching maximal expression at 24 h. ABA treatment (Figure 1F) induced a biphasic expression pattern in *GmbZIP59*, with initial gradual upregulation followed by significant downregulation after 12 h.

To validate the stress-responsive expression pattern of *GmbZIP59*, we generated a *ProGmbZIP59::GUS* transcriptional fusion construct and transformed it into *Arabidopsis* (Col-0) plants. Transgenic seeds were germinated and grown on standard 1/2 MS medium for four days. Uniform seedlings were then transferred to 1/2 MS medium supplemented with 150 mM NaCl, 250 mM mannitol, 350 μM ETH, 150 μM MeJA, or 1 μM ABA, and subjected to treatment durations of 0, 6, 12, and 24 h. The seedlings were collected and subjected to histochemical β-glucuronidase (GUS) staining after treatment. Seedlings were incubated overnight in GUS staining solution at 37 °C, followed by destaining with 75% (*v*/*v*) ethanol. The spatial and temporal expression dynamics of *GmbZIP59* under various stress conditions were visually documented (Figure 2).

GUS staining results indicated that under NaCl, mannitol, and ETH treatments, *GmbZIP59* expression peaked at 6 h post-treatment and subsequently declined. In response to MeJA, expression levels increased progressively with treatment duration, reaching a maximum at 24 h. Following ABA treatment, expression levels were similarly high at 6 and 12 h, after which they decreased. The GUS staining patterns corroborated the RT-qPCR data, confirming the conserved stress responsiveness of GmbZIP59 in heterologous systems.

### 2.3. Ectopic Expression of GmbZIP59 Enhances Resistance Against S. sclerotiorum in Arabidopsis

To elucidate the biological role of *GmbZIP59*, a *35S::GmbZIP59::GFP* construct was generated and transformed into *Arabidopsis*. Two independent transgenic lines, *OE-13* and *OE-20*, were selected for detached-leaf inoculation assays with *S. sclerotiorum*, using WT plants as controls (Appendix A). After 24 h of infection, significantly reduced lesion areas in *OE-13* and *OE-20* leaves were observed compared to WT (Figure 3A,C).

Further analysis using 3,3′-diaminobenzidine (DAB) staining demonstrated that *OE-GmbZIP59* plants accumulated lower levels of reactive oxygen species (ROS) than WT plants during infection (Figure 3B). These results indicate that *GmbZIP59* positively regulates plant defense response against *S. sclerotiorum* by modulating ROS homeostasis and limiting pathogen spread.

### 2.4. Ectopic Expression of GmbZIP59 Confers Salt Stress Tolerance but Enhances Phytohormone Sensitivity in Arabidopsis

Previous studies demonstrated that the expression of the *GmbZIP59* gene is induced by multiple stress conditions (Figure 1 and Figure 2). To further verify whether *GmbZIP59* participates in plant stress response processes, two *OE-GmbZIP59* transgenic *Arabidopsis* lines (*OE-13* and *OE-20*), empty vector, along with WT plants, were subjected to salt stress and three phytohormone treatments (MeJA, ETH, and ABA). Surface-sterilized seeds from both the transgenic and Col-0 were cultured on 1/2 MS medium supplemented with either 150 mM NaCl, 400 µM ETH, 150 µM MeJA, or 1 µM ABA for 7 days, after which phenotypic characteristics of each line were examined.

The results showed that under NaCl treatment, while transgenic *Arabidopsis* plants carrying empty vectors exhibit a growth phenotype similar to that of WT, the WT exhibited significantly weaker growth compared to the *OE-GmbZIP59* transgenic *Arabidopsis* (Appendix A and Figure 4A). Further measurements of root growth under NaCl treatment revealed that the root length of the WT was significantly shorter than that of the *OE-GmbZIP59* transgenic line (Appendix A and Figure 4B). Additionally, fresh weight measurements showed that the WT plants exhibited lower biomass than *OE-GmbZIP59* transgenic plants (Figure 4C).

In contrast, under treatments with 400 µM ETH, 150 µM MeJA, and 1 µM ABA, the *OE-GmbZIP59* transgenic *Arabidopsis* displayed significantly weaker growth compared to both the WT and empty vector controls, which exhibited similar growth phenotypes to each other (Appendix A and Figure 4A). Quantification of root length under these treatments confirmed that the *OE-GmbZIP59* transgenic line displays significantly shorter roots than WT (Appendix A and Figure 4B). Furthermore, fresh weight analysis under these stress conditions demonstrated that the *OE-GmbZIP59* transgenic plants had a lower fresh weight than the WT controls (Figure 4C). These results indicate a distinct physiological trade-off orchestrated by *GmbZIP59*, promoting survival under ionic stress (salt) while potentially intensifying defense responses to specific hormonal signals, even when those responses incur a growth penalty.

### 2.5. Ectopic Expression of GmbZIP59 Confers Salt and Drought Stress Tolerance in Rice

Expression assays revealed that the *GmbZIP59* gene responds to drought stress (Figure 1C). However, phenotypic analysis in *OE-GmbZIP59* transgenic *Arabidopsis* did not demonstrate a functional role in drought resistance. Given that rice is susceptible to drought stress due to its unique growth conditions, we generated a *GmbZIP59*-overexpressing rice line further to investigate its potential role in drought stress response. Using rice genetic transformation techniques, we introduced the overexpression vector (*35S::GmbZIP59::GFP*) into rice and obtained positive transgenic lines overexpressing *GmbZIP59* (*OE-GmbZIP59*). Two independent overexpression lines, *OE-1* and *OE-2*, were selected for subsequent analysis (Appendix A). This study utilized the T_2_ generation of transgenic rice for preliminary functional characterization.

Seeds of *OE-1*, *OE-2*, empty vector-containing transgenic controls, and the wild-type Zhonghua 11 (ZH11) were subjected to salt and drought stress treatments. Under control conditions, all lines exhibited comparable bud growth. After four days of treatment, phenotypic observations revealed that the empty vector controls and ZH11 displayed indistinguishable growth phenotypes under the conditions. However, under 150 mM NaCl treatment, ZH11 and empty vector seeds exhibited minimal germination, whereas the *OE-GmbZIP59* transgenic rice lines displayed robust bud growth. Similarly, under 250 mM mannitol-induced drought stress, ZH11 and empty vector seeds germinated but failed to develop shoots, in contrast to the *OE-GmbZIP59* transgenic rice lines, which exhibited significant shoot elongation (Appendix A and Figure 5A,C).

For salt and drought stress assays, 15-day-old hydroponically cultivated seedlings of *OE-GmbZIP59* transgenic rice (*OE-1*) and ZH11 were subjected to 10 days of treatment before phenotypic evaluation. The results showed that *OE-1* exhibited significantly enhanced growth performance compared to ZH11 under both stress conditions (Figure 5B). Leaf and root length measurements demonstrated differences between ZH11 and *OE-GmbZIP59* transgenic lines (Figure 5D,E).

### 2.6. Expression Analysis of Stress-Responsive Genes in OE-GmbZIP59 Transgenic Arabidopsis and Rice Under Biotic and Abiotic Stresses

To investigate the molecular mechanisms of *GmbZIP59*-mediated disease resistance and salt stress tolerance, we analyzed the expression of defense-related genes (*AtPR1*, *AtPAD3*, *AtACS4*, *AtLOX4*, *AtICS1*, and *AtPDF1.2*) and salt-responsive genes (*AtCOR6*-*6*, *AtSTZ*, *AtWRKY26*, *AtRD29B*, *AtABI5*, and *AtABA2*) in transgenic *Arabidopsis* lines using RT-qPCR with three technical replicates (triplicates). Leaf samples from Col-0, *OE-13*, and *OE-20* plants were collected at 24 and 48 h post-inoculation with *S. sclerotiorum* or following salt stress treatment.

Following RNA extraction and reverse transcription, we quantified the expression of these stress-responsive marker genes. Comparative analysis revealed significant upregulation of all tested genes in *OE-13* and *OE-20* transgenic plants compared to the WT controls (Figure 6). Among these, the transcript levels of *AtACS6*, *AtLOX4*, *AtICS1*, *AtPDF1.2*, *AtCOR6-6*, *AtSTZ*, *AtWRKY26*, *AtRD29B*, and *AtABI5* rose markedly over time and peaked at 48 h. *AtPR1*, *AtPAD3*, and *AtABA2* expression declined slightly after 24 h but remained higher than that in the WT. Statistical significance was determined using one-way ANOVA, with a power analysis confirming adequate sample size to detect significant differences. This analysis confirming the differential expression between transgenic and WT lines. These findings demonstrate that overexpression of *GmbZIP59* enhances *Arabidopsis* resistance to *S. sclerotiorum* infection (Figure 6A–F) and improves salt stress tolerance (Figure 6G–L) by activating defense-related and stress-responsive pathways.

To validate the expression pattern of stress-related genes in transgenic rice, leaf samples were collected from *GmbZIP59*-overexpressing lines and ZH11 plants 6, 12, 24, and 48 h after salt and drought stress exposure. RT-qPCR analysis of stress-responsive genes with three technical replicates (triplicates) revealed significantly higher expression levels in transgenic lines than WT controls at all examined time points. The expression levels of *OsDREB2A*, *OsDREB2B*, and *OsRD2A* increased significantly under both salt (Figure 7A–D) and drought stress (Figure 7E–H) conditions within 24 h but decreased by 48 h. Under salt stress, the expression level of *OsLEA3* showed a marked increase at 24 h, with only a slight rise by 48 h (Figure 7D). In contrast, under drought stress, the expression of *OsLEA3* peaked at 48 h (Figure 7H). Power analysis ensured the reliability of these statistical comparisons. These molecular findings were consistent with observed stress-tolerant phenotypes, confirming the functional role of *GmbZIP59* in enhancing stress resistance in rice.

### 2.7. Identification of GmbZIP59 Target Genes

To explore the molecular pathways through which *GmbZIP59* regulates stress tolerance, Chromatin immunoprecipitation (ChIP) was conducted to identify potential target genes of GmbZIP59-*35S::GmbZIP59::GFP* transient expression in two-week-old William 82 soybean plants. We designed the primers at both ends of the *cis*-acting element G-box of the relevant gene promoter. ChIP-qPCR detected the expression levels of biotic and abiotic stress-related genes. According to the quantitative results, among the binding amounts of the genes related to biological stress, the quantity of binding at the first binding site of *GmERF7* was the most significant (Figure 8A). At the same time, the GmbZIP59 protein can also bind to the *cis*-acting elements on the promoters of *GmPR2*, *GmRD22*, and *GmETR1* (Figure 8A). Among the non-biological stress-related genes, the binding quantity of GmbZIP59 to the second binding site of *GmDREB1B* was the most significant. At the same time, GmbZIP59 can bind to the *cis*-acting elements of the *GmABI5*, *GmBIP*, *GmERD1*, and *GmEIN2* promoters (Figure 8B).

Notably, these genes are associated with various hormone signaling pathways: *GmABI5*, *GmDREB1B*, and *GmRD22* are related to ABA signaling [22,23,24], *GmETR1*, *GmERF7*, and *GmEIN2* are associated with ETH signaling, *GmRD22* is related to JA signaling, and *GmPR2* is involved in SA signaling. These findings suggest that overexpression of *GmbZIP59* enhances biotic and abiotic stresses by increasing the transcription of genes involved in biotic and abiotic stress and multiple hormone signaling pathways.

## 3. Discussion

Soybean, as a crucial oil and food crop, is severely constrained by various biotic and abiotic stresses. Previous studies have indicated that bZIP transcription factors play key regulatory roles in plant responses to environmental stresses [25,26,27]. In this study, through systematic analysis of the soybean bZIP transcription factor family, we identified that the expression of *GmbZIP59* was most significantly altered under waterlogging and drought treatments [21]. We subsequently cloned the gene, constructed overexpression vectors, and introduced them into *Arabidopsis* and rice for functional characterization.

Phylogenetic analysis showed that GmbZIP59 shares high homology with stress-related bZIP proteins such as GmbZIP60 in soybean, AtbZIP44 and AtbZIP53 in *Arabidopsis*, and OsbZIP38 in rice (Appendix A). Previous studies have demonstrated that GmbZIP60 plays a significant role in enhancing salt and drought stress tolerance in soybean [19]. In *Arabidopsis*, AtbZIP44 has been implicated in the response to iron deficiency stress [28], while AtbZIP53 is involved in regulating adaptive responses to salinity and nutrient starvation [29,30]. In rice, OsbZIP38 interacts with OsOBF1 to form a heterodimer that contributes to the regulation of cold stress responses [31]. Meanwhile, expression profiling demonstrated that *GmbZIP59* is responsive to salt and drought stresses, and multiple phytohormones, including ETH, MeJA, and ABA-this result indicates its broad participation in abiotic stress adaptation (Figure 1 and Figure 2).

Ectopic expression of *GmbZIP59* in *Arabidopsis* significantly enhanced resistance to *S. sclerotiorum*, accompanied by reduced lesion size and ROS accumulation (Figure 3). This is consistent with previous reports that bZIP TFs can modulate plant immunity through regulating ROS homeostasis and defense gene expression [18,32]. Notably, transgenic plants exhibited increased sensitivity to exogenous ABA, MeJA, and ETH (Figure 4), indicating that *GmbZIP59* may potentiate hormone signaling to activate defense responses. Similarly, several bZIP TFs have been shown to interact with ABA, MeJA, and ETH signaling pathways to enhance biotic stress resistance [26,31].

Under salt stress, *GmbZIP59*-overexpressing *Arabidopsis* plants showed improved growth performance, longer roots, and higher fresh weight compared to WT (Figure 4), demonstrating its positive role in salt tolerance. However, unlike its homolog *GmbZIP60*, which enhances drought tolerance in both *Arabidopsis* and rice [19], *GmbZIP59* did not confer drought tolerance in *Arabidopsis*. This functional divergence may be attributed to structural differences or distinct regulatory mechanisms among bZIP members. Interestingly, transgenic rice plants overexpressing *GmbZIP59* exhibited improved tolerance to salt and drought stresses (Figure 5), highlighting the species-specific functionality of this TF. Rice, being more sensitive to drought due to its growth environment, may utilize *GmbZIP59* in adaptive mechanisms that are not fully operational in *Arabidopsis*.

Mechanistically, GmbZIP59 likely exerts its protective effects by directly binding to promoters of stress-responsive genes and modulating their expression. Our ChIP-qPCR analysis confirmed that GmbZIP59 binds to cis-elements in the promoters of key genes involved in ABA (*GmABI5*, *GmDREB1B*, *GmRD22*), ETH (*GmETR1*, *GmERF7*, *GmEIN2*), and JA (*GmRD22*) signaling (Figure 8). Accordingly, qRT-PCR analyses revealed that transgenic plants exhibited upregulated expression of multiple stress-responsive markers, including *AtPR1*, *AtPAD3*, *AtACS6*, *AtLOX4* in *Arabidopsis* and *OsDREB2A*, *OsRD29A*, *OsLEA3* in rice (Figure 6 and Figure 7), supporting the notion that *GmbZIP59* enhances stress tolerance through coordinated activation of defense-related pathways.

Based on the in silico promoter analysis of *Arabidopsis* target genes using the EPD database, we identified that AtbZIP44 and AtbZIP53—the *Arabidopsis* orthologs of soybean GmbZIP59—bind to the promoter regions of multiple stress- and hormone-related genes (such as *AtRD22*, *AtETR1*, *AtNPR3*, *AtBIP*, *AtDREB1B*, *AtERD1*, and *AtEIN2*) with high confidence (*p*-value = 0.001). Notably, several binding sites are located near the transcription start site (TSS), including positions −19 in *AtERF7* and −71 in *AtETR1*, suggesting a direct role in regulating transcription initiation (Appendix A). The prevalence of binding sites in the upstream promoter regions supports the potential of these bZIP transcription factors to act as transcriptional regulators, likely influencing the expression of genes involved in stress response and hormone signaling.

These findings imply that *GmbZIP59* may function similarly in soybean, potentially regulating a conserved set of target genes involved in abiotic stress responses and hormone pathways, such as ETH, ABA, and MeJA. The conserved binding capability across species underscores the functional importance of *GmbZIP59* in modulating transcriptional programs critical for plant adaptation to environmental challenges.

In conclusion, our results demonstrate that *GmbZIP59* functions as a multifunctional transcription factor in plant stress responses. In *Arabidopsis*, it enhances resistance to fungal pathogens and salt stress. In rice, it confers tolerance to both salt and drought stress. These effects are likely mediated through the integration of ETH, JA, and ABA signaling and activating downstream defense genes. Our study demonstrates that *GmbZIP59* plays a critical role in regulating plant stress resistance, highlighting its potential as a valuable genetic resource for improving stress resilience in crops. It should be noted, however, that our validation was conducted under controlled laboratory conditions and that the application of transgenic technology in agriculture faces regulatory restrictions. Therefore, translating this finding into practical applications will require further investigation into non-transgenic strategies and robust field validation. Future research should focus on defining its direct regulatory targets via ChIP-Seq and interactome analyses, employing CRISPR-Cas9 to create targeted mutations or promoter modifications in the endogenous *GmbZIP59* locus of model crops to evaluate the phenotypic outcome, and conducting multi-location field trials with engineered or selected lines to assess their agricultural value comprehensively.

## 4. Materials and Methods

### 4.1. GmbZIP59 Gene Isolation and Vector Construction

Based on the coding sequence (CDS) and promoter region of the soybean GmbZIP59 gene (Glyma. 10G013300) obtained from Phytozome (https://phytozome-next.jgi.doe.gov accessed on 1 January 2023), the required fragments were synthesized for subsequent over-expression and GUS-reporter constructs.

The full-length CDS was amplified by PCR using the specific primers GmbZIP59-F (5′-CACCATGGCTTCGATTCC-3′) and GmbZIP59-R (5′-AGCAAACAGGCCTTGATTTCCATGG-3′) (Appendix A). The amplification procedure was as follows: 95 °C for 30 s (pre-denaturation); followed by 40 cycles of 94 °C for 5 s (denaturation), 60 °C for 15 s (annealing), 72 °C for 10 s (extension). The resulting PCR product was then cloned into the GATEWAY entry vector pENTR-GmbZIP59. An LR recombination reaction between pENTR-GmbZIP59 and the destination vector pGWB605 generated the final over-expression vector, designated OE-GmbZIP59. To elucidate the biological role of *GmbZIP59*, we generated an overexpression construct (*35S::GmbZIP59::GFP*) and transformed it into *Arabidopsis*.

### 4.2. Plant Growth Conditions

*Arabidopsis thaliana plants* ecotype Columbia-0 (Col-0) were cultivated on a soil mixture [2:1 (*v*/*v*) nutrient soil (Pindstrup, Denmark): vermiculite] in plastic pots in an innovative greenhouse. The plants were fertilized by soil irrigation with a 600-diluted solution of compound fertilizer (Hetaili, Beijing, China) every 15 days. Growth conditions were maintained at 22 °C, 65% relative humidity, with a 16 h light/8 h dark photoperiod provided by LED panels and a light intensity of 120 µmol m^−2^ s^−2^. Thirty-day-old mature plants were harvested for transformation.

For rice, the variety used was Zhonghua 11 (ZH11), which was grown in loam soil with an appropriate amount of compound fertilizer added. Plants were cultivated under 28–30 °C, 50% relative humidity, and a 14 h light/10 h dark photoperiod supplied with LED growth lamps, and a light intensity of 500 µmol m^−2^ s^−2^. Seeds were collected at maturity for callus induction and subsequent transformation.

### 4.3. Pathogens and Inoculation Procedures

For the inoculation with *S. sclerotiorum*, the fungal strains preserved at 4 °C were first subcultured on potato dextrose agar medium for two days. Subsequently, a 7 mm puncher was used to obtain fresh hyphae from the actively growing margins of the colonies. These strains were then closely upended onto the leaves of three-week-old plants. The inoculated leaves were placed in a square Petri dish and transferred to a growth chamber conducive to disease development, where they were incubated overnight. Lesion areas were quantified using ImageJ 1.47 software [33]. All experiments were performed with three independent replicates.

### 4.4. Stress Tolerance Assays and Measurements of Physiological Indices

Surface-sterilized seeds of both Col-0 and transgenic lines were spotted onto either plain or treatment-supplemented 1/2 Murashige and Skoog Medium (MS). This was followed by incubation in a greenhouse for 7 days, after which root length and fresh weight were measured. The Col-0 and transgenic *Arabidopsis* plants grown on the medium were then transferred to potting soil and cultivated for 13 days. Different concentrations of treatment solutions were subsequently applied, and plant height was recorded after an additional 15 days of growth.

Zhonghua 11 (ZH11) and overexpression rice seeds were germinated in Petri dishes with a gibberellin solution for 24 h. After treatment with various concentrations of the test solutions, the seeds were transferred to a rice greenhouse, and shoot length was measured after 4 days of cultivation. Germinated seeds were also placed in tissue culture boxes containing an adequate volume of complete rice nutrient solution and grown in the rice greenhouse. Upon reaching the two-leaves-one-heart stage, the nutrient solution was replaced with fresh medium supplemented with different concentrations of the treatment solutions. After 15 days of stress treatment, leaf length was assessed. All experiments were performed with three biological replicates.

### 4.5. RNA Extraction and Quantitative Real-Time PCR

To isolate total RNA from the leaves of William 82 (*Glycine max*), we employed an RNA extraction kit (Omega Bio-Tek, Shanghai, China). Subsequently, cDNA synthesis was performed using PrimerScript™ RTase (TaKaRa Biotechnology, Beijing, China) according to the manufacturer’s protocol. The relevant primers are in Appendix A.

### 4.6. Histological Detection of GUS Activity in GmbZIP59

Sterilized seeds of *ProGmbZIP59::GUS* transgenic and wild-type *Arabidopsis* (Col-0) were germinated on Murashige and Skoog (MS) medium. After 4d, uniform seedlings were transferred to fresh MS plates containing one of the following treatments: 150 mM NaCl, 250 mM mannitol, 350 μM ETH, 150 μM MeJA, and 1 μM ABA, or no supplement (control). Seedlings were grown vertically for an additional 24 h before histochemical GUS staining. They were incubated at 37 °C overnight in X-Gluc staining buffer. Subsequently, chlorophyll was removed with 75% (*v*/*v*) ethanol before imaging.

### 4.7. Chromatin Immunoprecipitation (ChIP) Analysis

Chromatin immunoprecipitation (ChIP) was performed on 2-week-old William 82 leaves transiently overexpressing *35S::GmbZIP59::GFP*. Approximately 4 g of tissue was vacuum-infiltrated with 1% (*v*/*v*) formaldehyde for cross-linking. Nuclei were isolated and chromatin was digested with 0.2 U micrococcal nuclease (Sigma, St. Louis, MO, USA) in 1 mL MNase buffer (10 mM Tris-HCl, pH8.0; 50 mM NaCl; 1 mM β-mercaptoethanol; 0.1% NP-40; 1 mM CaCl_2_; and 1×cOmplete™ protease inhibitor cocktail, Roche) for 10 min at 37 °C. Digestion was terminated by adding 5 mM EDTA. After centrifugation, the soluble chromatin was immunoprecipitated overnight at 4 °C with anti-GFP antibody (Abcam, Cambridge, UK). DNA–protein complexes were recovered with protein A/G magnetic beads, washed, and eluted. Cross-links were reversed at 65 °C for 6 h, and DNA was purified using a PCR cleanup kit. Enrichment of target fragments was quantified by qPCR on a CFX96 system (Bio-Rad, Berkeley, CA, USA) using gene-specific primers (Appendix A). All ChIP assays were performed in duplicate; representative data are shown.

### 4.8. Promoter Binding Site Analysis for bZIP Transcription Factors

To complement the experimental ChIP-qPCR data and strengthen the mechanistic insight into *GmbZIP59* function, an in silico promoter analysis was performed. The promoter sequences (typically defined as 1500 bp upstream of the transcription start site, TSS) of the identified *Arabidopsis* target genes (e.g., *AtRD22*, *AtETR1*, *AtNPR3*, *AtEIN2*) were retrieved from The *Arabidopsis* Information Resource (TAIR, https://www.arabidopsis.org/ accessed on 1 February 2023). These sequences were subsequently analyzed using the Eukaryotic Promoter Database (EPD, https://epd.expasy.org/epd/ accessed on 1 October 2025) and relevant plant transcription factor binding motif databases to predict the presence of bZIP-type binding motifs (e.g., G-box, ABRE). The position of each predicted binding site relative to the TSS was precisely determined. High-confidence binding events for the *Arabidopsis* bZIP orthologs, AtbZIP44 and AtbZIP53, were identified based on a significance threshold of *p*-value ≤ 0.001, as provided by the EPD.

## Figures and Tables

**Figure 1 plants-14-03326-f001:**
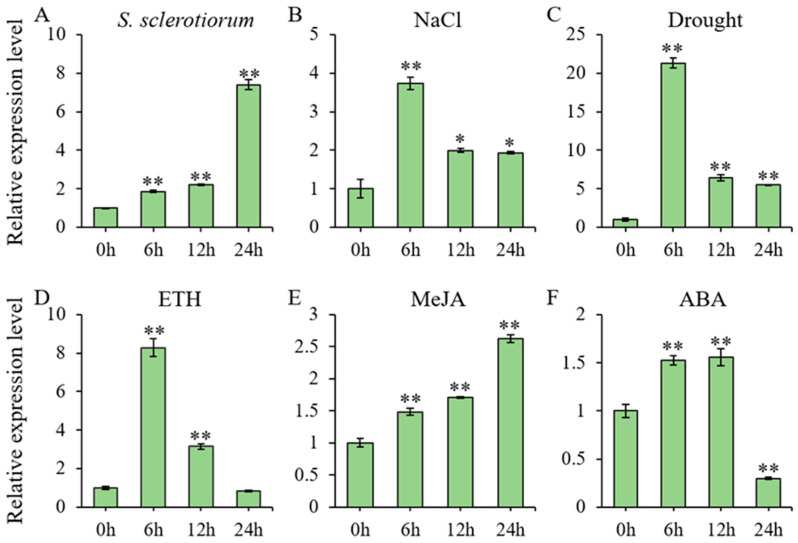
Stress-responsive expression profiles of *GmbZIP59* in soybean. (**A**) Temporal expression patterns of *GmbZIP59* in WT plants following *S. sclerotiorum* infection, quantified by RT-qPCR. (**B**–**F**) *GmbZIP59* transcript dynamics in response to: (**B**) 150 mM NaCl (salt stress), (**C**) 250 mM mannitol (drought stress), (**D**) 350 μM ETH, (**E**) 150 μM MeJA, and (**F**) 1 μM ABA. Two-week-old soybean (William 82) seedlings were used for all treatments. Error bars represent the mean ± SD of three biological replicates. Significant differences relative to control (0 h) are indicated by asterisks (** *p* < 0.01, 0.01 < * *p* < 0.05; Student’s *t*-test).

**Figure 2 plants-14-03326-f002:**
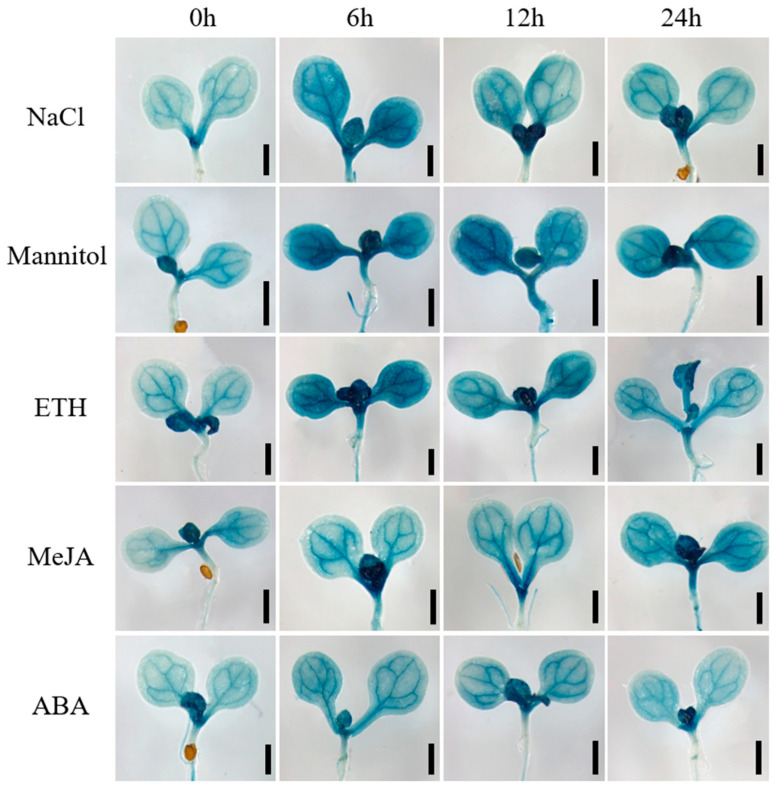
The *GmbZIP59* promoter is activated by NaCl, mannitol, ETH, MeJA, and ABA. Histochemical analysis of GUS activity in *ProGmbZIP59::GUS* transgenic *Arabidopsis* seedlings revealed that promoter activity was strongly induced by NaCl, mannitol, ETH, MeJA, and ABA treatments, as visualized by intense blue staining in leaves. Scale bar = 1 mm.

**Figure 3 plants-14-03326-f003:**
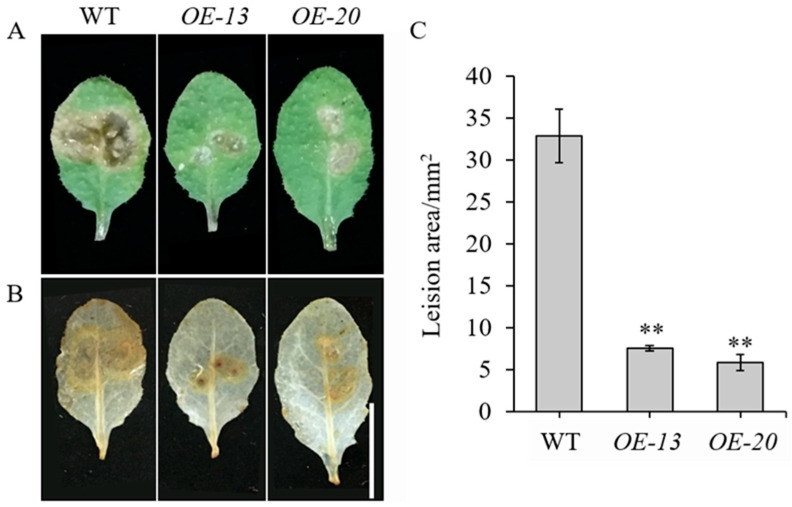
Overexpression of *GmbZIP59* enhances disease resistance of *Arabidopsis* to *S. sclerotiorum* infection (**A**) Representative images of detached leaves from WT and *OE-GmbZIP59* transgenic lines (*OE-13* and *OE-20*) at 24 h post-inoculation. (**B**) ROS accumulation visualized by DAB staining. Bar = 1 cm. (**C**) Measurement of lesion areas showing enhanced resistance in transgenic lines. Error bars represent the mean ± SD (*n* = 3 biological replicates). Significant differences versus WT controls are indicated by asterisks (** *p* < 0.01; Student’s *t*-test).

**Figure 4 plants-14-03326-f004:**
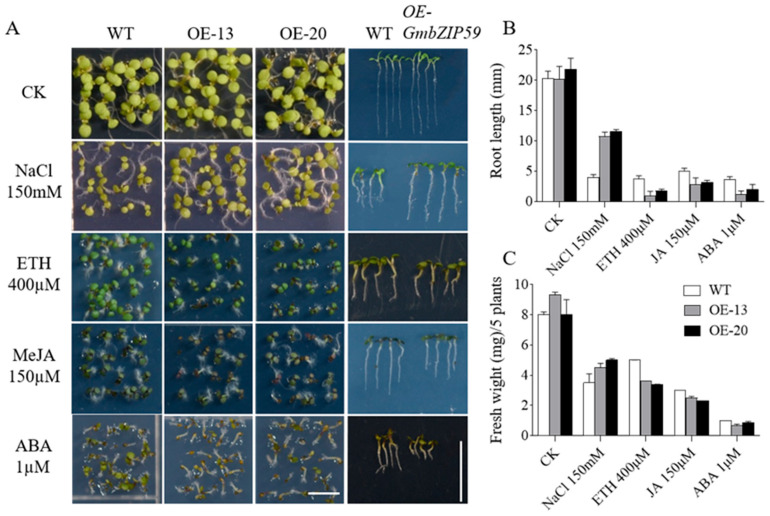
Phenotypic analysis of *GmbZIP59* transgenic *Arabidopsis* plants under salt and plant hormone treatment. (**A**) All the seeds were germinated on the 1/2 Murashige and Skoog Medium (MS) medium under control conditions (CK) or supplemented with 150 mM NaCl, 400 µM ETH, 150 µM MeJA, and 1.0 µM ABA for 7 days, Bar = 0.5 cm. (**B**) Calculation of root length, Bar = 1 cm. (**C**) Fresh weight measurements of seedlings. Two independent *OE-GmbZIP59* transgenic rice lines (*OE-13* and *OE-20*) were analyzed.

**Figure 5 plants-14-03326-f005:**
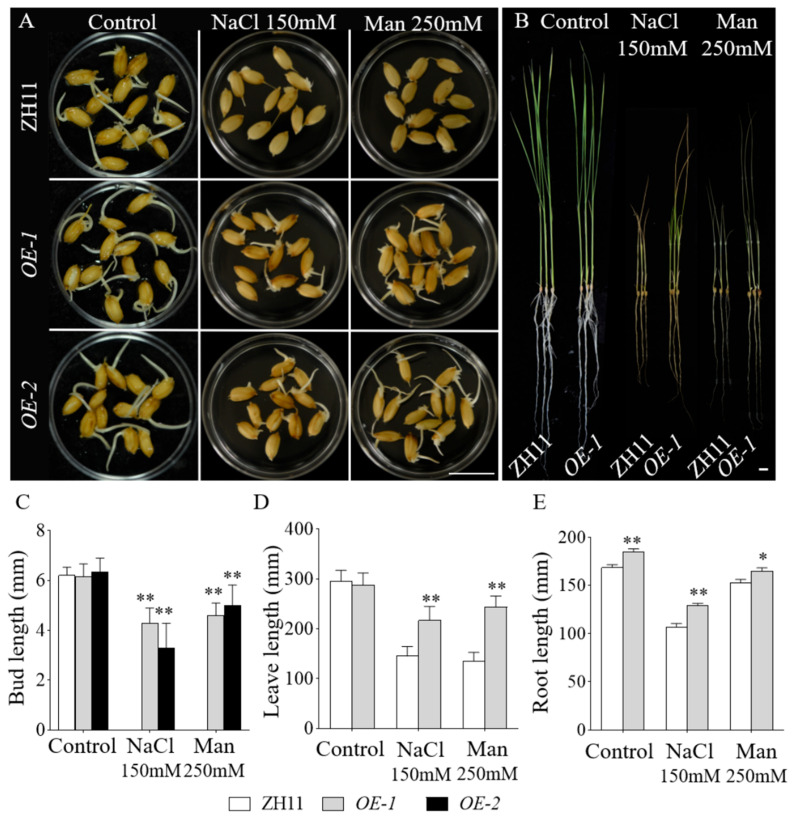
*OE-GmbZIP59*-overexpressed rice lines exhibited enhanced tolerance to salt and drought compared to WT. (**A**) Seeds of *OE-1*, *OE-2*, and ZH11 (wild-type) were germinated under control conditions or subjected to NaCl (150 mM) and mannitol (250 mM) treatments for 4 days. Scale bar, 1 cm. (**B**) *OE-1* and ZH11 (wild-type) were germinated under control conditions or exposed to 150 mM NaCl and 250 mM mannitol for 10 days. Scale bar, 1 cm. (**C**) Measurement of the seedling bud length. (**D**) Measurement of the plant leaf length. (**E**) Measurement of the plant root length. Error bars represent the mean ± SD of three biological replicates. Asterisks denote significant differences between the indicated comparisons based on Student’s *t*-test (** *p* < 0.01; 0.01 < * *p* < 0.05).

**Figure 6 plants-14-03326-f006:**
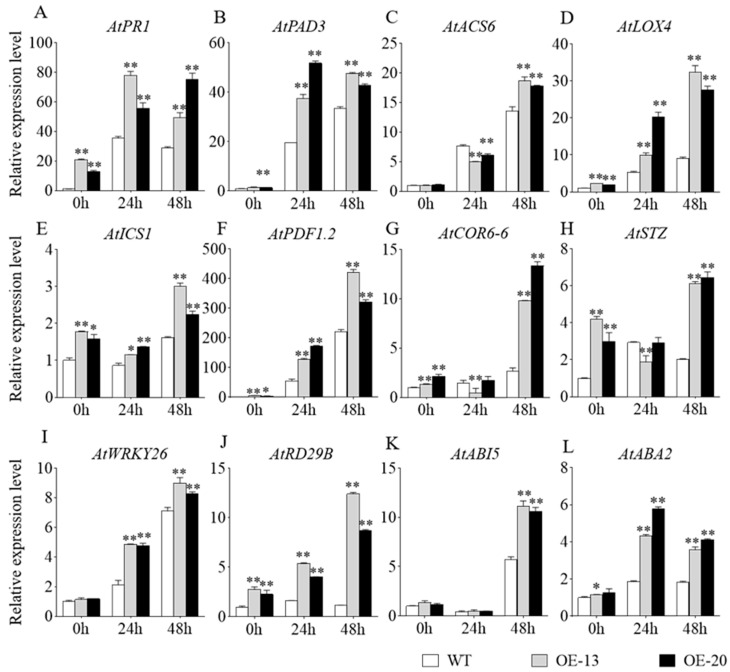
Expression profiles of defense-related and abiotic stress-responsive genes in the WT and *OE-GmbZIP59* transgenic *Arabidopsis* plants following *S. sclerotiorum* inoculation and salt stress treatment. (**A**–**F**) Relative expression levels of defense-related genes (*AtPR1*, *AtPAD3*, *AtACS6*, *AtLOX4*, *AtICS1*, and *AtPDF1*.*2*). (**G**–**L**) Relative expression levels of abiotic stress-responsive genes (*AtCOR6*-*6*, *AtSTZ*, *AtWRKY26*, *AtRD29B*, *AtABI5*, and *AtABA2*). Error bars represent the mean ± SD (*n* = 3 biological replicates with three technical replicates each). Data were analyzed by one-way ANOVA. Asterisks denote statistically significant differences determined by Student’s *t*-test (** *p* < 0.01; 0.01 < * *p* < 0.05). Complete statistical analysis including power calculation is provided in Appendix A.

**Figure 7 plants-14-03326-f007:**
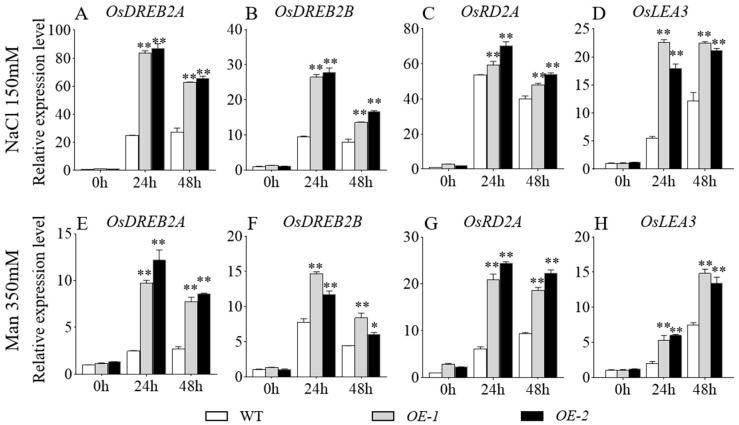
The expression of abiotic stress-related genes in the WT and *OE-GmbZIP59* transgenic rice plants in response to salt (**A**–**D**) and drought (**E**–**H**) stresses. Error bars represent the mean ± SD (*n* = 3 biological replicates with three technical replicates each). Data were analyzed by one-way ANOVA. Asterisks denote statistically significant differences determined by Student’s *t*-test (** *p* < 0.01; 0.01 < * *p* < 0.05). Complete statistical analysis including power calculations is provided in Appendix A.

**Figure 8 plants-14-03326-f008:**
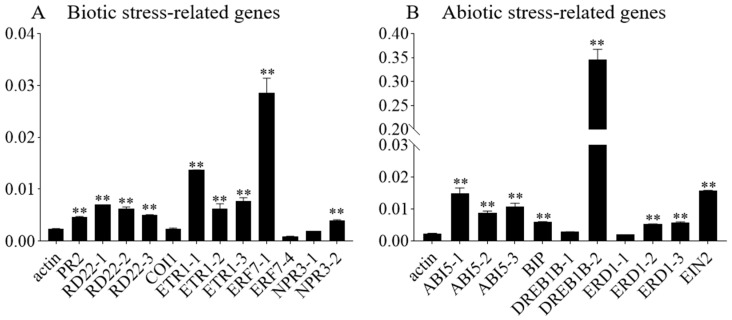
The chromatin immunoprecipitation (ChIP) result of *OE-GmbZIP59* transiently expressing in soybean. ChIP-qPCR analysis of GmbZIP59 binding to promoters of abiotic stress-related genes using GFP antibody and *OE-GmbZIP59* transgenic soybean plants. Error bars indicate ± SD of three biological replicates. Asterisks indicate significant differences for the indicated comparisons based on Student’s *t*-test (** *p* < 0.01).

## Data Availability

The original contributions presented in this study are included in the article/Appendix A. Further inquiries can be directed to the corresponding authors.

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
