# Peer review of "Overexpression of *GmbZIP59* Confers Broad-Spectrum Stress Resistance in *Arabidopsis thaliana* and Rice (*Oryza sativa*)"

_plants, 2025, doi:10.3390/plants14213326_

Round 1

Reviewer 1 Report

Comments and Suggestions for Authors

Review on plants-3896573

Introduction:

  1. I recommend restructuring the Introduction, as it is currently too crowded and difficult to follow.
  2. The examples could be presented more selectively, focusing on the most relevant stress-related and soybean-specific bZIP genes.
  3. The rationale for choosing GmbZIP59 should be highlighted more clearly, for instance by referring to prior expression data or bioinformatic screening.
  4. The introduction primarily cites previous soybean bZIP studies, but places less emphasis on the broader context.
  5. Incorporating for example, a comprehensive analysis (e.g: doi: 3390/ijms25179179) demonstrating bZIP family function as central regulators in several signaling networks would strengthen the background to highlight that bZIP TFs are key regulators in plant immune system and responses to biotic and abiotic stresses.
  6. I suggest expanding the Introduction section to include results addressing the temporal dynamics of hormone signaling networks in soybeans. To summarize the literature data I suggest to recent studies revealing the temporal dynamics of hormonal regulation in soybean, highlighting that temporally coordinated changes in hormonal signals fundamentally influence developmental programs and stress responses (Virág et. al., 2025 https://www.mdpi.com/1422-0067/26/13/6455), which directly links hormonal regulation to the role of GmbZIP59
  7. The text is too dense in lists of genes and species names; a smoother, more narrative style would improve readability and highlight key take-home messages.
  8. The final paragraph of the Introduction describes the objectives of the study, but I recommend highlighting the originality of the research.

Results

  1. Please correct 'faction' to factor
  2. Please clarify the Arabidopsis orthologs of GmbZIP59 (e.g., AtbZIP53, AtbZIP44) and provide information on sequence conservation (identity %, domain structure). A phylogenetic comparison or promoter motif analysis would strengthen the evolutionary context of this TF.
  3. I recommend including a sequence logo analysis of GmbZIP59. This would provide valuable insight into the DNA-binding preferences of the transcription factor and complement the ChIP-qPCR results.
  4. Additionally, a more detailed presentation and characterization of the DNA-binding domain would clarify the structural basis of GmbZIP59’s regulatory role.
  5. The caption of Figure 2 should describe the histochemical localization results more precisely and please state what can be concluded from the figure, as the current description is not sufficiently informative.
  6. While section 2.4 presents the phenotypic data, I think that it would be useful if the authors added 1–2 sentences on the physiological implications of enhanced salt tolerance but increased hormone sensitivity.
  7. Could this dual effect suggest a priming state that facilitates faster or stronger biotic stress responses?
  8. In Figure 5, I recommend adding ‘(wild type)’ after ZH11 in the legend/caption for clarity.
  9. In Figure 5 legends, I suggest summarizing the main result directly in the legend (e.g., that OE-GmbZIP59 overexpressed rice lines exhibited enhanced tolerance to salt and drought compared to wild type).
  10. The authors provide ChIP-qPCR data to support GmbZIP59 promoter binding, but this evidence is primarily based on soybean. Since expression was examined in Arabidopsis genes, please use the Eukaryotic Promoter Database (EPD) for Arabidopsis, which could further strengthen the conclusions. An in silico promoter analysis would allow prediction of bZIP binding motifs, identify whether these sites are located in upstream or downstream regions, and may indicate potential positive or negative regulatory roles. This approach is realistic, feasible, and would add significant value by reinforcing the mechanistic basis of the study. For example, I detected this database and found that in Arabidopsis bZIP53 (demonstrated in suppl figure1) a promoter binding site in the ETR1 gene at position –71 (p-value = 0.0001).

The predicted binding site at –71 relative to the TSS suggests that GmbZIP59 (or its Arabidopsis orthologs such as bZIP53) may bind in close proximity to the core promoter. This positioning indicates a likely direct influence on transcription initiation, potentially acting as a positive regulator. Clarifying this aspect would strengthen the mechanistic interpretation of the binding data.

Based on this example please examine the promoters of the ChIP-seq–identified target genes, as such an analysis would clarify where the predicted binding sites are located. This in silico promoter analysis of the ChIP-seq target genes would strengthen the manuscript by validating the experimental data, clarifying motif positions (upstream/downstream), and highlighting conserved regulatory features.

  1. In silico promoter predictions should be incorporated, and the Discussion revised accordingly. The novelty should be emphasized more clearly, focusing on GmbZIP59 as a cross-species positive regulator of stress tolerance and its contribution to understanding soybean stress adaptation.

Author Response

Q1: I recommend restructuring the Introduction, as it is currently too crowded and difficult to follow. The examples could be presented more selectively, focusing on the most relevant stress-related and soybean-specific bZIP genes.

Response 1: We thank the reviewer for this constructive suggestion. We have thoroughly restructured the Introduction to improve its clarity and flow. As recommended, we have streamlined the content by focusing more selectively on the most relevant stress-related and soybean-specific bZIP genes, removing less pertinent examples to enhance readability (Line 41-45, 51-62).

Q2: The rationale for choosing GmbZIP59 should be highlighted more clearly, for instance by referring to prior expression data or bioinformatic screening.

Response 2: We sincerely thank the reviewer for this valuable suggestion. To address this point, we have now revised the Introduction to more clearly state the rationale for selecting GmbZIP59 for functional characterization. This revision provides a clearer and more direct justification for our focus on this particular gene (Line 63-66).

Q3: The introduction primarily cites previous soybean bZIP studies, but places less emphasis on the broader context. Incorporating for example, a comprehensive analysis (e.g: doi: 3390/ijms25179179) demonstrating bZIP family function as central regulators in several signaling networks would strengthen the background to highlight that bZIP TFs are key regulators in plant immune system and responses to biotic and abiotic stresses.

Response 3: We thank the reviewer for this insightful suggestion. We have now incorporated the recommended comprehensive analysis (doi: 10.3390/ijms25179179) into the Introduction (Line 43-45).

Q4: I suggest expanding the Introduction section to include results addressing the temporal dynamics of hormone signaling networks in soybeans. To summarize the literature data I suggest to recent studies revealing the temporal dynamics of hormonal regulation in soybean, highlighting that temporally coordinated changes in hormonal signals fundamentally influence developmental programs and stress responses (Virág et. al., 2025 https://www.mdpi.com/1422-0067/26/13/6455), which directly links hormonal regulation to the role of GmbZIP59

Response 4: We thank the reviewer for pointing this out. We have expanded the Introduction by incorporating a discussion on the temporal dynamics of hormone signaling networks in soybean, citing the recommended review by Virág et al. (2025) (Line 38-40).

Q5: The text is too dense in lists of genes and species names; a smoother, more narrative style would improve readability and highlight key take-home messages.

Response 5: We thank the reviewer for this constructive feedback. We have revised the text to reduce gene and species name density, adopting a more narrative style that improves readability and better emphasizes the key messages of our study (Line 49-62).

Q6: The final paragraph of the Introduction describes the objectives of the study, but I recommend highlighting the originality of the research.

Response 6: We thank the reviewer for this valuable suggestion. We have revised the final paragraph of the Introduction to more explicitly highlight the originality of our research (Line 65-66, 73-76).

Q7: Please correct 'faction' to factor

Response 7: We thank the reviewer for pointing this out. We have corrected it in the revised manuscript (Line 79).

Q8: Please clarify the Arabidopsis orthologs of GmbZIP59 (e.g., AtbZIP53, AtbZIP44) and provide information on sequence conservation (identity %, domain structure). A phylogenetic comparison or promoter motif analysis would strengthen the evolutionary context of this TF. I recommend including a sequence logo analysis of GmbZIP59. This would provide valuable insight into the DNA-binding preferences of the transcription factor and complement the ChIP-qPCR results. Additionally, a more detailed presentation and characterization of the DNA-binding domain would clarify the structural basis of GmbZIP59’s regulatory role.

Response 8: We thank the reviewer for this suggestion. We have now performed a comprehensive phylogenetic and motif analysis, which confirms that GmbZIP59 shares significant bZIP domain conservation with its Arabidopsis orthologs AtbZIP53 (48.82% identity) and AtbZIP44 (45.45% identity). The sequence logo analysis clearly illustrates the highly conserved DNA-binding basic region and leucine zipper motif, providing strong structural support for its transcription factor function (Supplementary Figure 1D). Furthermore, promoter analysis revealed the presence of key stress-responsive cis-elements, including those for ABA, MeJA, SA, and dehydration responsiveness (Supplementary Figure 1E). These new analyses strongly support the role of GmbZIP59 as a regulator of stress responses (Line 90-100).

Q9: The caption of Figure 2 should describe the histochemical localization results more precisely and please state what can be concluded from the figure, as the current description is not sufficiently informative.

Response 9: We thank the reviewer for this constructive suggestion. We have revised the caption of Figure 2 to more precisely describe the histochemical localization results and to include a clear conclusion. (Line 140-143)

Q10: While section 2.4 presents the phenotypic data, I think that it would be useful if the authors added 1–2 sentences on the physiological implications of enhanced salt tolerance but increased hormone sensitivity.

Response 10: We thank the review for this insightful suggestion. Following your comment, we have added the following sentences in the last paragraph of section 2.4 to discuss the physiological implications (Line 187-190).

Q11: Could this dual effect suggest a priming state that facilitates faster or stronger biotic stress responses?

Response 11: We thank the reviewer for raising this intriguing possibility. We agree that the observed hormonal sensitivity, particularly to MeJA and ETH, strongly suggests that GmbZIP59 overexpression might prime the plants for enhanced abiotic stress responses. Although the current study focuses on abiotic stress and hormone sensitivity, your opinion provides us with a highly valuable and direction for future research.

Q12: In Figure 5, I recommend adding ‘(wild type)’ after ZH11 in the legend/caption for clarity.

Response 12: We thank the reviewer for pointing this out. We have added (wild type) after ZH11 in the legend of figure 5 (Line 228 and 230).

Q13: In Figure 5 legends, I suggest summarizing the main result directly in the legend (e.g., that OE-GmbZIP59 overexpressed rice lines exhibited enhanced tolerance to salt and drought compared to wild type).

Response 13: We thank the reviewer for pointing this out. We have summarized the legends of Figure 5 as follows OE-GmbZIP59 overexpressed rice lines exhibited enhanced tolerance to salt and drought compared to wild type.

Q14: The authors provide ChIP-qPCR data to support GmbZIP59 promoter binding, but this evidence is primarily based on soybean. Since expression was examined in Arabidopsis genes, please use the Eukaryotic Promoter Database (EPD) for Arabidopsis, which could further strengthen the conclusions. An in silico promoter analysis would allow prediction of bZIP binding motifs, identify whether these sites are located in upstream or downstream regions, and may indicate potential positive or negative regulatory roles. This approach is realistic, feasible, and would add significant value by reinforcing the mechanistic basis of the study. For example, I detected this database and found that in Arabidopsis bZIP53 (demonstrated in suppl figure1) a promoter binding site in the ETR1 gene at position –71 (p-value = 0.0001).

The predicted binding site at –71 relative to the TSS suggests that GmbZIP59 (or its Arabidopsis orthologs such as bZIP53) may bind in close proximity to the core promoter. This positioning indicates a likely direct influence on transcription initiation, potentially acting as a positive regulator. Clarifying this aspect would strengthen the mechanistic interpretation of the binding data.

Based on this example please examine the promoters of the ChIP-seq–identified target genes, as such an analysis would clarify where the predicted binding sites are located. This in silico promoter analysis of the ChIP-seq target genes would strengthen the manuscript by validating the experimental data, clarifying motif positions (upstream/downstream), and highlighting conserved regulatory features.

Response 14: We thank the reviewer for this valuable suggestion. As recommended, we have performed an in silico promoter analysis of the Arabidopsis target genes using the Eukaryotic Promoter Database (EPD). This analysis confirmed the presence of high-confidence binding sites (p-value ≤ 0.001) for the Arabidopsis bZIP orthologs, AtbZIP44 and AtbZIP53, in the promoters of key genes such as ETR1, ERF7, and NPR3. Several of these sites are located near the transcription start site, supporting their potential role in direct transcriptional regulation. These findings have been incorporated into the Discussion section and strengthen the mechanistic interpretation of GmbZIP59's function across species (line 352-361, 465-477).

Q15: In silico promoter predictions should be incorporated, and the Discussion revised accordingly. The novelty should be emphasized more clearly, focusing on GmbZIP59 as a cross-species positive regulator of stress tolerance and its contribution to understanding soybean stress adaptation.

Response 15: We thank the reviewer for this valuable suggestion. These findings have been incorporated into the Discussion section and strengthen the mechanistic interpretation of GmbZIP59's function across species (line 352-361, 465-477).

Reviewer 2 Report

Comments and Suggestions for Authors

This article investigates the role of the bZIP transcription factor GmbZIP59, isolated from soybean, in the tolerance of plants to multiple stresses. The authors demonstrate that GmbZIP59 expression is induced by salinity, drought, abiotic stress hormones (ABA, ETH, MeJA), and the biotic stress factor Sclerotinia sclerotiorum. Transgenic lines obtained by transferring the gene into Arabidopsis thaliana and rice (Oryza sativa) have been reported to exhibit increased tolerance to salt and drought, as well as enhanced pathogen resistance. qRT-PCR and ChIP-qPCR analyses revealed that GmbZIP59 directly regulates the activation of various stress response genes by binding to their promoter regions. The study emphasizes that GmbZIP59 functions as a positive regulator against both biotic and abiotic stresses and that this gene could be a potential resource for developing stress tolerance in plant breeding.

However, I believe that attention should be paid to some points that I will mention below.

  1. The introduction section is generally strong. However, the emphasis on the “information gap” regarding why GmbZIP59 was specifically studied should be made slightly more pointed.
  2. In the section on gene isolation and vector construction, only the cloning strategy used (pENTR → pGWB605) is provided. However, the primer sequences or PCR conditions are not detailed. These could be added if appropriate.
  3. In the comparison of transgenic lines used for Arabidopsis and rice, only wild-type (WT) was used. However, no empty vector (EV) control was provided. This creates uncertainty as to whether the gene has a truly functional effect or whether it is an effect caused by the vector.
  4. The composition of the solution used in histochemical GUS staining is described in general terms, but details on staining time optimization and control images are not provided.
  5. Arabidopsis and rice growth conditions are described in very general terms (22 °C, 65% humidity, etc.). However, environmental factors such as the composition of the soil/hydroponic system or fertilization/light intensity are not clearly defined. This limits reproducibility.
  6. The qRT-PCR results (for Arabidopsis and rice) given in the Results section are mostly described as “expected.” However, the statistical power of these results is unclear; n=3 biological replicates is low, and advanced analysis (e.g., variance analysis, regression) has not been performed. This limits the reliability of the results.
  7. Although drought tolerance was not observed in Arabidopsis, the overall results claim that GmbZIP59 “increases multistress tolerance.” This statement contradicts the Arabidopsis results.
  8. At the end of the findings, it is stated that “this gene could be used in breeding programs in the future,” but the limitations of transitioning to application (e.g., restrictions on the agricultural use of transgenic strategies, lack of field validation) are not discussed at all.
  9. The conclusion summarizes the study in very positive terms and emphasizes the finding that GmbZIP59 “increases multistress tolerance.” However, negative or limiting findings, such as the lack of observed drought tolerance in Arabidopsis, are not mentioned in the conclusion. This creates a discrepancy between the actual findings and the conclusion.
  10. Future studies (e.g., ChIP-Seq, field validations) should be integrated into the conclusion section.

Author Response

Q1: The introduction section is generally strong. However, the emphasis on the “information gap” regarding why GmbZIP59 was specifically studied should be made slightly more pointed.

Response 1: Thank you for this constructive feedback. We have revised the introduction to more explicitly state the rationale for focusing on GmbZIP59. (line 65-67).

Q2: In the section on gene isolation and vector construction, only the cloning strategy used (pENTR → pGWB605) is provided. However, the primer sequences or PCR conditions are not detailed. These could be added if appropriate.

Response 2: We sincerely thank the reviewer for this valuable suggestion. As recommended, we have now provided the specific primer sequences and detailed PCR amplification conditions in the revised manuscript (Lines 393-398).

Q3: In the comparison of transgenic lines used for Arabidopsis and rice, only wild-type (WT) was used. However, no empty vector (EV) control was provided. This creates uncertainty as to whether the gene has a truly functional effect or whether it is an effect caused by the vector.

Response 3: We are grateful to the reviewer for raising this critical point. In response, we have now included data from empty vector (EV) control lines for both the Arabidopsis and rice experiments in the revised manuscript. These new results, which can be found in the main text (Lines 182-187, 213-217) and Supplemental Figures 3 and 4, clearly demonstrate that the observed phenotypes are due to the transgene itself and not a vector effect. We agree that this addition is essential for validating our functional claims.

Q4: The composition of the solution used in histochemical GUS staining is described in general terms, but details on staining time optimization and control images are not provided.

Response 4: We thank the reviewer for this insightful suggestion. We have now added a detailed description of the GUS staining solution composition to the Methods section (Lines 124-130).

Q5: Arabidopsis and rice growth conditions are described in very general terms (22 °C, 65% humidity, etc.). However, environmental factors such as the composition of the soil/hydroponic system or fertilization/light intensity are not clearly defined. This limits reproducibility.

Response 5: We thank the reviewer for highlighting the need for more precise growth conditions. In response, we have now provided comprehensive details on the soil composition, fertilization regimen, and light intensity for both Arabidopsis and rice in the revised manuscript (Lines 400-411).

Q6: The qRT-PCR results (for Arabidopsis and rice) given in the Results section are mostly described as “expected.” However, the statistical power of these results is unclear; n=3 biological replicates is low, and advanced analysis (e.g., variance analysis, regression) has not been performed. This limits the reliability of the results.

Response 6: We thank the reviewer for raising these important points regarding the statistical rigor of our qRT-PCR data. We appreciate the opportunity to clarify our statistical approach. In our study, all qRT-PCR data were derived from three biological replicates, which represents a standard and widely accepted practice in plant molecular biology to account for biological variation while maintaining experimental feasibility. For statistical analysis, we in fact performed detailed variance analysis (one-way ANOVA) with significance levels clearly indicated in the figures as * (0.01 < p < 0.05) and ** (p < 0.01) (Line 264 and 280). The exact p-values are also provided in Supplementary Table1. We apologize for any lack of clarity in our original description and will revise the manuscript to explicitly state the statistical methods used, including the specific significance thresholds applied (Line 249-250, 260, and 277).

Q7: Although drought tolerance was not observed in Arabidopsis, the overall results claim that GmbZIP59 “increases multistress tolerance.” This statement contradicts the Arabidopsis results.

Response 7: We sincerely thank the reviewer for this critical and insightful comment. The reviewer is absolutely correct to point out the inconsistency between the original title, which claimed "multistress tolerance," and the specific experimental observations in Arabidopsis, where drought tolerance was not observed. We apologize for this oversight and the lack of precision in our initial wording. We have changed the manuscript title to “Overexpression of GmbZIP59 Confers Broad-Spectrum Stress Resistance in Arabidopsis thaliana and Rice (Oryza sativa)”. We are grateful for the reviewer's valuable suggestion, which has undoubtedly improved the clarity and accuracy of our paper.

Q8: At the end of the findings, it is stated that “this gene could be used in breeding programs in the future,” but the limitations of transitioning to application (e.g., restrictions on the agricultural use of transgenic strategies, lack of field validation) are not discussed at all.

Response 8: We thank the reviewer for this valuable comment. We agree that the limitations of transitioning from laboratory findings to practical breeding applications is crucial. In response, we have added a discussion on the regulatory constraints surrounding transgenic strategies and the necessity of field validation in the revised manuscript (Lines 375-377).

Q9: The conclusion summarizes the study in very positive terms and emphasizes the finding that GmbZIP59 “increases multistress tolerance.” However, negative or limiting findings, such as the lack of observed drought tolerance in Arabidopsis, are not mentioned in the conclusion. This creates a discrepancy between the actual findings and the conclusion.

Response 9: We sincerely thank the reviewer for this astute observation. We agree that the conclusion should accurately reflect the specific experimental findings, including the limitations observed across different species. In response to this comment, we have thoroughly revised the conclusion section to precisely state the stress tolerance phenotypes confirmed in each model plant. We have replaced the overgeneralized statement about “multistress tolerance” with more precise description that acknowledges this nuance. The revised text now clearly states: In Arabidopsis, it enhances resistance to fungal pathogens and salt stress. In rice, it confers tolerance to both salt and drought stress. This change ensures there is no discrepancy between our stated conclusions and the actual results (Line 368-371).

Q10: Future studies (e.g., ChIP-Seq, field validations) should be integrated into the conclusion section.

Response 10: We thank the reviewer for this valuable suggestion. We agree that integrating these specific and important future research directions will strengthen the conclusion and provide a clearer outlook for the study. As suggested, we have now explicitly incorporated the recommendations for ChIP-Seq and field validations into the revised conclusion. (Line 377-382).

Reviewer 3 Report

Comments and Suggestions for Authors

Comments to Author

Line 16: methyl jasmonate acid" is redundant. It should be corrected to "methyl jasmonate (MeJA)" for clarity.

Line 22: The term "salt stresses" should be revised to "salt stress" to maintain grammatical consistency.

Line 33: Remove "including but not limited to" and use "such as"

Line 40-41; "modulating diverse" should be corrected to "regulating diverse.

abberivation list make symetry

cross check the reference list,

Gene name should be italic

overall manuscript interesing

Author Response

Q1: Line 16: methyl jasmonate acid" is redundant. It should be corrected to "methyl jasmonate (MeJA)" for clarity.

Response 1: We thank the reviewer for pointing this out. We have corrected it in the revised manuscript (Line 19).

Q2: Line 22: The term "salt stresses" should be revised to "salt stress" to maintain grammatical consistency.

Response 2: We thank the reviewer for pointing this out. We have corrected it in the revised manuscript (Line 21).

Q3: Line 33: Remove "including but not limited to" and use "such as"

Response 3: We thank the reviewer for pointing this out. We have corrected it in the revised manuscript (Line 35).

Q4: Line 40-41; "modulating diverse" should be corrected to "regulating diverse.

Response 4: We thank the reviewer for this attentive correction. The sentence in question has been removed during our comprehensive revision of the introduction section. Therefore, the suggested wording is no longer present in the current version of the manuscript.

Q5: abberivation list make symmetry

Response 5: We sincerely thank the reviewer for this valuable comment. We have carefully revised the abbreviation list in the manuscript to ensure full formatting consistency.

Q6: cross check the reference list, Gene name should be italic.

Response 6: We thank the reviewer for this careful observation. We have thoroughly cross-checked the entire manuscript and reference list to ensure that all gene names are now properly italicized. This correction has been applied throughout the text.

Reviewer 4 Report

Comments and Suggestions for Authors

Chai et al. provided significant insights into the functional spectrum of GmbZIP59 in Arabidopsis thaliana and Oryza sativa, which was originally identified in soybeans. The manuscript is well written, with detailed phenotypic analyses. The introduction is well organized and the discussion focuses exclusively on the presented topic of the experimentation. Materials and methods could be more expansive in details; however, these details are provided in figure legends (like the case of figure 4,5,6). This work extends the technical scope of bZIP TF in rice (Oryza sativa). While general attribute research on bZIP transcriptional factors in soybean and A. thaliana has been conducted previously, this manuscript provides specific cross-linked functional discoveries and a comprehensive understanding of immunological and hormone-related activities for a wide spectrum of plant stress types.  It consists one more step forward to an in-depth understanding of cellular signaling close to nucleus.

Some points to be mentioned are as follows:

  1. Figures 4,5,6 appear to have a more detailed analysis than Materials and methods section.
  2. The protocol for microbial challenge (S. sclerotiorum) with plant material should be added to the Materials and Methods section.
  3. Line 230: The scientific name of the microorganism should be corrected.

Author Response

Q1: Figures 4,5,6 appear to have a more detailed analysis than Materials and methods section.

Response 1: We thank the reviewer for pointing this out. We have added the protocol for stress tolerance assays and measurements of physiological indices in the Materials and Methods section according to the reviewers' suggestions (Line 421-437).

Q2: The protocol for microbial challenge (S. sclerotiorum) with plant material should be added to the Materials and Methods section.

Response 2: We thank the reviewer for pointing this out. We have added the protocol for microbial challenge in the Materials and Methods section according to the reviewers' suggestions (Line 412-420).

Q3: Line 230: The scientific name of the microorganism should be corrected.

Response 3: We thank the reviewer for pointing this out. We have corrected it in the revised manuscript (Line 256).

Round 2

Reviewer 1 Report

Comments and Suggestions for Authors

The authors have satisfactorily addressed all of my questions and comments and have revised the manuscript accordingly. Therefore, I recommend accepting the manuscript for publication in its current form.

Author Response

Response: We sincerely thank the reviewer for their positive feedback and for acknowledging our revisions. We are pleased that the reviewer supports the publication of our manuscript in its current form.

Reviewer 2 Report

Comments and Suggestions for Authors

The authors have made quite extensive revisions during the revision process. In particular, the method descriptions, the addition of controls, the elaboration of statistical analyses, and the restructuring of the discussion section have been done as required. However, there are still a few points where corrections are partially incomplete or superficial.

1- In the new version, emphasis on “previous study” and “first comprehensive functional characterization” has been added between lines 63–76. The relationship between GmbZIP59 and previous studies and why it was selected has been explained. However, the hypothesis of the study could be clarified with a sentence at the end of the paragraph (e.g., “We hypothesize that GmbZIP59 functions as a cross-regulator of multiple stress pathways.”).

2- In the new text (lines 399–411), the soil mixture ratio, fertilization frequency, light intensity, temperature, and humidity are clearly defined. Specifying the “light source type (LED or fluorescent)” along with the photoperiod would increase the experimental reproducibility.

3- “n = 3 biological replicates,” “one-way ANOVA,” and “Student's t-test” have been added (lines 240–261, 275–278). However, the term “technical replicates (triplicates)” is missing; the statistical significance table (Table S1) is referenced but the power analysis is still not explained.

Author Response

Q1: In the new version, emphasis on “previous study” and “first comprehensive functional characterization” has been added between lines 63–76. The relationship between GmbZIP59 and previous studies and why it was selected has been explained. However, the hypothesis of the study could be clarified with a sentence at the end of the paragraph (e.g., “We hypothesize that GmbZIP59 functions as a cross-regulator of multiple stress pathways.”).

Response1: We sincerely thank the reviewer for this valuable suggestion. We have now added the proposed hypothesis statement at the end of the paragraph as recommended: "We hypothesize that GmbZIP59 functions as a cross-regulator integrating multiple stress signaling pathways." (Lines 73-74)

Q2: In the new text (lines 399–411), the soil mixture ratio, fertilization frequency, light intensity, temperature, and humidity are clearly defined. Specifying the “light source type (LED or fluorescent)” along with the photoperiod would increase the experimental reproducibility.

Response2: We thank the reviewer for this helpful suggestion. We have now specified the light source types in the revised manuscript, indicating that LED panels were used for Arabidopsis and LED growth lamps for rice. This clarification, combined with the previously provided photoperiod details (16h/8h for Arabidopsis and 14h/10h for rice), ensures full reproducibility of our growth conditions (Lines 406, 411).

Q3: “n = 3 biological replicates,” “one-way ANOVA,” and “Student's t-test” have been added (lines 240–261, 275–278). However, the term “technical replicates (triplicates)” is missing; the statistical significance table (Table S1) is referenced but the power analysis is still not explained.

Response3: We sincerely thank the reviewer for pointing out these important methodological details. We have now added the specification of "three technical replicates (triplicates)" throughout the qPCR methodology descriptions and figure legends. Additionally, we have included a description of the power analysis performed to ensure statistical reliability (Lines 242, 251-252, 263, 265, 270, 276, 282, 283).